# CoSMIC: A hybrid approach for large-scale, high-resolution microbial profiling of novel niches

Maor Knafo[1], Shahar Rezenman[1], Tal Idan[1], Michael Elgart[2], Shlomi Dagan[3], Vered Zavaro[1], Ziv Reich[1], Ruti Kapon[1], Dagan Sade[1]*, Noam Shental[4]*

1 Department of Biomolecular Sciences, Weizmann Institute of Science, Rehovot, Israel, 2 Metabolomic Discovery Unit, Metabolic Center, Sheba Medical Center, Ramat Gan, Israel, 3 Fisher Drug Discovery Resource Center, The Rockefeller University, New York, New York, United States of America, 4 Department of Mathematics and Computer Science, The Open University of Israel, Ra'anana, Israel

she These authors contributed equally to this work.

* dagan.sade@weizmann.ac.il (DS); shental@openu.ac.il (NS)

## Abstract

Standard microbial profiling based on 16S rRNA (16S) sequencing suffers from a lack of primer universality, primer biases, and often yields low resolution. We introduce 'Comprehensive Small Ribosomal Subunit Mapping and Identification of Communities' (CoSMIC), addressing these challenges, especially in unexplored niches. CoSMIC begins with long-read sequencing of the full-length 16S gene, amplified by generic Locked Nucleic Acid primers over pooled samples, thus augmenting reference databases with novel niche-specific gene sequences. Subsequently, CoSMIC amplifies multiple non-consecutive variable regions along the gene, followed by short-read sequencing of each sample. Data from the different regions are integrated using the SMURF framework, alleviating primer biases and providing de facto full gene resolution. Using a mock community, CoSMIC identified full-length 16S genes with significantly higher specificity and sensitivity while dramatically increasing resolution compared to standard methods. Evaluating CoSMIC across environmental samples yielded higher accuracy and unparalleled resolution at a fraction of the cost of standard long-read sequencing per sample while allowing the detection of thousands of novel full-length 16S sequences.

## Introduction

During the last decade, with the advent of massively parallel sequencing, the study and exploration of microbiomes have become an integrative science that spans multiple disciplines and research fields. Medicine, environmental studies, agriculture, and even behavioral studies have associated specific phenotypes with bacteria [1–4]. Most of these studies rely on sequencing predefined variable regions of the 16S rRNA gene using short reads. However, it has been widely shown that the results of such studies depend on the amplified region due to biases introduced

**Data availability statement:** Raw reads used in this study can be obtained from the NIH SRA database BioProject ID: PRJNA1172999 -https://www.ncbi.nlm.nih.gov/bioproject/1172999. Code and supporting data are available at https://github.com/NoamShental/CoSMIC.

**Funding:** Funding was provided by Sustainability and Energy Research Initiative by TI; The Yotam project and the Weizmann Institute sustainability and energy research initiative to DS; Combined Weizmann-Abroad Postdoctoral Grant for Advancing Women in Science, the Emergency Fellowships for Israeli Postdocs Affected by COVID-19 and the European Union's Horizon Europe research and innovation programme under the Marie Skłodowska-Curie grant agreement No 101106697 (HOLOSPONGIA) to TI. The Open University of Israel internal grant 10024 to NS. The funders had no role in study design, data collection and analysis, decision to publish, or preparation of the manuscript.

**Competing interests:** The authors have declared that no competing interests exist.

by the specific primers applied [5,6], hence limiting the ability to harmonize findings across studies [7]. Moreover, due to short read length, standard methods amplify one or two adjacent variable regions; therefore, many bacteria may share their sequence over the amplicon yet be very different along the rest of the 16S rRNA gene [7–9]. Such ambiguity in assigning reads to bacteria often results in lower resolution profiling. However, current bioinformatic pipelines do not adequately address either problem, regardless of whether they use amplicon sequence variants (ASVs) or operational taxonomic units (OTUs) [7,8].

To overcome these problems, we have previously presented the Short MUlti-Region Framework (SMURF) [10] approach. In SMURF, each sample is amplified by any number of primer pairs, and reads from different regions are computationally combined to provide coherent, high-resolution microbial profiling. For such integration, SMURF relies on available databases of full-length 16S rRNA sequences (e.g., SILVA [11] or Greengenes [12]) containing millions of sequences. These databases are almost comprehensive with respect to human-associated bacteria, while other habitats may be severely underrepresented [13,14], rendering the approach suboptimal for less explored niches.

Here, we present Comprehensive 16S rRNA gene Mapping and Identification of Communities (CoSMIC), which builds on SMURF to provide a framework for high-resolution profiling of novel niches. Before applying SMURF in a new niche, we suggest amplifying a pool of samples using generic locked nucleic acid (LNA) primers [15], which provide an almost universal amplification of full-length 16S rRNA sequences. Sequencing these amplicons by long-read technologies (e.g., PacBio) enriches 16S rRNA gene databases with relevant sequences, thus allowing SMURF to provide efficient and high-resolution profiling of these niches. We provide a workflow for applying CoSMIC detailing the selection of specific subsets of primers to enable cost savings while not compromising resolution. The advantages of CoSMIC over standard methods are demonstrated by sequencing a mock mixture and over a variety of environmental samples.

CoSMIC is best suited for profiling novel ecological niches. It provides de facto full-length 16S rRNA resolution at about 10% of the cost of per-sample PacBio sequencing. This cost-effectiveness, combined with CoSMIC's flexibility in primer selection, allows researchers to optimize their experimental designs for both scope and budget. In addition, the full-length 16S rRNA sequences detected in CoSMIC's initial step may be uploaded to public 16S rRNA databases, thus allowing other researchers to perform even more cost-effective profiling studies over these niches.

## Materials and methods

### Sample collection and preservation

Overall, collected samples consisted of 33 individual plants and their surroundings, one mock community sample, two sample pools derived from multiple plants and their surroundings (collected in distinct locations in Israel), two sample pools derived from sponges collected in Israel and France, and another pool composed of all 33 individual samples used in the study (S1 Table). Leaf, root, and soil samples were

collected in the field and kept in a nucleic acid protection buffer [16]. Sponge samples were collected and immediately preserved in absolute ethanol. Upon arrival at the lab, samples were kept at -80°C until DNA extraction.

### DNA extraction

Genomic DNA was extracted from 100mg of the root and leaf samples, 250mg of soil samples, and an approximately 1cm x 0.5cm piece from the sponges using the ZymoBIOMICS DNA Kit (Cat. D4300), following the manufacturer's protocol.

For the mock community, DNA was extracted from each isolate using GenElute Bacterial Genomic DNA Kits (Cat. NA2110-1KT), and 20ng from each species was used to create a mock community.

Pools were created by adding 50ng from each amplicon post LNA amplification and cleaning up to generate a 500ng pool.

The following subsections describe the specific samples and all steps per processing pipeline (the order of the presentation corresponds to the Results sections).

### Field collection permits

All samples were collected in accordance with local and national regulations. Israel: Permits for collecting plant samples were granted by the Israel Nature and Parks Authority (permit numbers: 2021/42189, 2021/42190). Sponge samples from Israel's coast (off Herzliya, 32.17710°N, 34.63306°E) were collected under permit number 2021/42929, also issued by the Israel Nature and Parks Authority. Voucher specimens of the collected Israeli sponge material have been deposited in the Steinhardt Museum of Natural History at Tel Aviv University. No specific permits were required for collecting soil samples, which complies with Israeli regulations for microbial sampling from public lands.

France: Sponge samples were collected within the Calanques National Park (near Marseille, 43°12'38.0"N 5°20'12.3"E). These collections were authorized under permit Arrêté N° 47 DU 29 JANVIER 2020, granted to the Institut Pythéas for scientific sampling by the Préfet de la Région Provence-Alpes-Côte d'Azur.

### The Qiagen processing pipeline as used in Figs 1 and 2

**Samples.** Leaf, soil, and root samples of the following plants were collected: *A. poliothamnus*, *B. hancei*, *T. mongolica*, *L. sativus*, and *A. hirsutiflora*. In addition, two *L. apetalum* plants were collected. In total, 24 samples were processed.

**Library preparation and sequencing.** Libraries for deep sequencing were prepared using the QIAseq 16S/ITS Region Panel protocol (Cat. 333842), which amplifies the V1V2, V2V3, V3V4, V4V5, V5V7, and V7V9 regions (primer sequences may be available upon request from Qiagen). Sequencing was performed on an Illumina MiSeq using 250—8—8—250 cycles. No-template controls (NTC) were included in each library preparation and sequencing run. An analysis of these controls is provided in the Supplementary Results, confirming the absence of significant contamination in our profiling results.

**Qiagen-based analysis in Fig 1.** The resulting FASTQ files were analyzed using the QIAGEN CLC Genomics Workbench software with the CLC Microbial Genomics module, which performs closed-reference Operational Taxonomic Unit (OTU) analysis. More specifically, the software preprocesses the raw sequencing data, aligns the reads against a reference database (Greengenes [12] clustered at 99%), and clusters the sequences into OTUs based on a similarity threshold of 97%. It then assigns taxonomy to each OTU and generates an OTU table containing the frequency of each OTU in each sample.

**Combinatorial estimation of sets of variable regions in Fig 2.** The same FASTQ files were provided to SMURF which was independently applied to the 63 combinations of the six amplified regions (Fig 2). The performance of each combination was evaluated by: (a) ambiguity - the number of sequences in each SMURF-detected group, i.e., in each

set of 16S rRNA genes having the same sequence over the amplified regions; (b) mutual species - the proportion of correctly identified species, i.e., the number of headers that appear in both groups (six regions-full, and region combination-partial) this parameter would be 1 if the partial combination has all the headers and 0 if they share none with the full six-region analysis; and (c) Pearson correlation between the correctly identified mutual species. A Jupyter Notebook tutorial is available (GitHub NoamShental/CoSMIC).

**Enriching the database with novel 16S rRNA gene sequences**

**Samples for pooled LNA amplification and PacBio long-read sequencing in Fig 3.** Four pools were created, each by combining extracted DNA from relevant samples:

*Spongia officinalis France pool*: Samples from seven different individuals were collected along the coast of Marseille, France.

*S. officinalis Israel pool*: Samples from three different individuals were collected along the coast of Haifa, Israel.

*Arava desert Israel pool*: Leaf, root and surrounding soil samples from the following Arava desert species were pooled: *Moringa oleifera* Lam., *Pulicaria incisa* DC, *Ochradenus baccatus* Delile, *Anabasis articulata* (Forssk.) Moq., *Salsola vermiculata* (L.) Akhani & Roalson, *Commiphora gileadensis* (L.) C.Chr., *Fagonia mollis* Delile, *Zilla spinosa* Prantl, and *Asteriscus graveolens* (Forssk.) Less.

*Temperate lowlands Israel pool*: Leaf, root, and surrounding soil samples from the following lowlands region species were collected near Rehovot, Israel: *Retama raetam* (Forssk.) Webb & Berthel., *Ferula communis* L., *Vicia villosa* Roth, *Heterotheca subaxillaris* (Lam.) Britton & Rusby, *Sarcopoterium spinosum* (L.) Spach, *Prasium majus* L. and *Withania somnifera* (L.) Dunal.

**Samples for pooled LNA amplification and Loop Genomics synthetic long reads sequencing in Fig 5.** A pool was created for LNA amplification and Loop Genomics synthetic long-read sequencing [17] by combining 11 different libraries: a mixture of the 24 samples prepared using the Qiagen QIAseq kit; one library per each of the nine samples prepared using the Swift 16S+ITS PANEL kit; and one library for the mock community.

**Designing LNA primers for full-length 16S rRNA sequences.** Using a refined version of the Greengenes database containing 1.4M full-length 16S rRNA gene sequences [12], we sought short primer pairs (of length 10-13nt) that maximize the number of amplified genes. Considering only perfect matches while ignoring thermodynamic considerations [15], we identified three candidate pairs that potentially amplify 1.2M out of the 1.4M sequences, corresponding to 86% of Greengenes sequences (analogously, these primers potentially amplify 84% of SILVA NR99 sequences). Candidate pairs were provided to Qiagen to replace specific nucleotides with locked nucleic acids, resulting in our LNA primer set (Table 1 and S17 Fig).

**Sample preparation.** The LNA primer set was applied to augment the database with novel full-length 16S rRNA gene sequences to be sequenced by either PacBio or Loop Genomics pipelines. The forward and reverse primers were combined into two separate mixes at a stock concentration of 20µM.

Samples were individually amplified at an input concentration of 10-50ng. Each sample was amplified using the following volumes: 25µl Kapa HiFi HotStart ReadyMix (cat. KK2602), 1.25µl forward primer pool, and 1.25µl reverse primer pool (each to a final concentration of 0.5µM), 50ng template, and DDW up to 50µl. PCR plan used: 95°C - 3 minutes, 2 cycles: 98°C - 20 seconds, 55°C - 15 seconds, 72°C - 1 minute; 24 cycles: 98°C - 20 seconds, 58°C - 15 seconds, 72°C - 1 minute; 72°C - 5 minutes, hold at 4°C. Following amplification, the products were cleaned using Ampure XP beads (Cat. A63881) at 0.6X concentration. The relevant samples from each niche were pooled at 100 ng/µl each (for a full list of samples for each pool, see S1 Table).

**PacBio sequencing.** 16S rRNA full-length amplicons obtained from LNA amplification of the microbial communities in *Spongia officinalis* (France and Israel) and in plants from Israel's lowland and Arava were used as starting templates for the SMRTbell Express Template Prep Kit 2.0 (Product PN. 100-938-900). Sequencing was performed using

**Table 1**. **LNA primers.** A. Three primer pairs (pairs 1-3) and their location along the *E. coli* 16S rRNA gene. Primers are available via the Qiagen catalog number. The specifics of modified bases are a Qiagen trade secret, yet only a handful of bases are modified. B. The number of amplified full-length 16S rRNA genes by each pair and by their intersection. About 360K are amplified by all three pairs, 570K by two pairs, and 250K by just a single pair.

**A. Primer Sequences**

| Primer | Sequence | Qiagen catalog number |
|---|---|---|
| fwd_pair1 (LNA-F-52) | CCACATGCAAG | YCO0190926 (339406) |
| rev_pair1 (LNA-R-1382) | GGAACGTATTCAC | YCO0190932 (339406) |
| fwd_pair2 (LNA-F-32) | AACGCTGGCG | YCO0190928 (339406) |
| rev_pair2 (LNA-R-1376) | CCTATTCACCG | YCO0190929 (339406) |
| fwd_pair3 (LNA-F-57) | CCGCAAGTCG | YCO0190930 (339406) |
| rev_pair3 (LNA-R-1406) | GACGGGCGGT | YCO0190931 (339406) |

**B. Amplification Counts**

| | pair1 | pair2 | pair3 |
|---|---|---|---|
| pair1 | 1,008,809 | 700,830 | 570,143 |
| pair2 | 700,830 | 751,633 | 379,021 |
| pair3 | 570,143 | 379,021 | 712,836 |

the PacBio Sequel platform [18]. Only circular consensus sequences (CCS) with >3 passes and quality scores >Q40 were retained, ensuring >99.5% per-base accuracy. The SMRT [18] analysis software performed additional quality filtering and chimera removal (reads are eliminated if two SMRTbell adaptors are sequenced). To address residual sequencing errors ( 0.5%), all sequences were clustered at 99% identity using CD-HIT (see "Calling novel 16S rRNA genes"), which merges sequences differing by minor base errors into single representatives. This combination of high-accuracy CCS reads and 99% clustering prevents sequencing errors from being mistaken for novel sequences.

**Loop genomics sequencing.** Samples used to validate the CoSMIC method (Fig 5) were sequenced using Loop Genomics [17]. 16S rRNA full-length amplicons were used as a template for the LoopSeq PCR Amplicon Kit. Sequencing was done on the Illumina NovaSeq sequencer using the following cycles 151—8—8—151, resulting in 2.5-17 Mbp per sample (S16 Fig). Analysis was performed on the company's remote server, yielding fasta files containing assembled long reads.

**Filtering potential chimeras.** Although the SMRT analysis software already filters out potential chimeric reads, and to support chimera detection for Loop Genomics reads, the uchime_denovo algorithm [19] was applied to detect and remove chimeric sequences. While no chimeric reads were found among full-length 16S rRNA sequences that were to be "augmented" to our database, we acknowledge the possibility that such sequences may still be present. The formation of chimeric sequences during 16S rRNA amplification, especially for full-length genes, has been well-documented in the literature [7]. Combining chimera-detection algorithms and the fact that CoSMIC's profiling is robust to chimeric sequences provides significant assurances regarding the validity of detected sequences.

**Calling novel 16S rRNA genes following the sequencing of LNA amplicons.** Candidate 16S rRNA gene sequences were clustered within each experiment using CD-HIT [20] at 99% identity to remove redundant sequences. Non-redundant sequences were compared to the SILVA database using the cd-hit-est-2d function to determine their alignment to SILVA at different identity levels (Fig 3A). Script for this analysis appears in the GitHub repository "NoamShental/CoSMIC". This script was used for sequences of both PacBio and Loop Genomics.

## The Swift processing pipeline as used in Fig 5

**Samples.** Leaf, soil, and root samples of *T. hirsuta*, *S. lycopersicum*, and *T. aestivum* were collected.

A mock community was formed using 12 species obtained from environmental isolates: *Acidovorax citrulli*, *Bacillus subtilis*, *Paenibacillus dendritiformis*, *Clavibacter michiganensis*, *Escherichia coli*, *Zymomonas mobilis*, *Proteus mirabilis*,

*Pseudomonas syringae*, *Xanthomonas campestris*, *Agrobacterium tumefaciens*, *Sphingomonas yanoikuyae*, and *Bacillus pumilus*. Species identification was performed by Sanger sequencing using E8f-939r as primers.

**Library preparation and sequencing.** Libraries for deep sequencing were prepared with the Swift AMPLICON™ 16S+ITS PANEL (Cat. AL-51696), using five forward and four reverse primers pooled in a single reaction (Table 2). Sequencing was performed on the Illumina NovaSeq sequencer with the following cycles: 151—8—8—151, yielding 2.5-15 Mbp reads per sample (S15 Fig lower panel). Base calling and demultiplexing were performed using Illumina's bcl2fastq command-line interface.

**Analyzing a commercial standard mock mixture.** To further validate the CoSMIC pipeline, we also analyzed a commercially available mock community, i.e., the ZymoBIOMICS Microbial Community DNA Standard (Zymo Research, Cat. No. D6305). The DNA standard was processed through the same multi-region short-read sequencing pipeline as our custom mock community. The resulting composition was compared with the manufacturer's theoretical specifications (S21 Fig) and showed high similarity to the prepared community.

**Data analysis.** *Standard analysis:* Microbial profiling was performed using the standard Swift analysis scripts [21] to create Amplicon Sequence Variant (ASV) or OTU tables. These tables were based on the QIIME2 [22] package, where paired R1 and R2 reads were filtered and analyzed using R's DADA2 [23] package to account for sequencing errors.

*CoSMIC analysis:* Each sample was demultiplexed to its variable regions using Cutadapt [24]. In principle, since five forward primers were mixed with four reverse primers, SMURF can be applied to up to 20 specific regions. We selected nine potential amplicons (Table 3) while ignoring long amplicons (e.g., the pair V1_f and V9_r was omitted) since the number of such reads was negligible. Paired R1 and R2 reads were filtered, analyzed using R's DADA2 package to account for sequencing errors, and submitted to SMURF.

## A practical guide to primer selection in CoSMIC

CoSMIC involves two distinct sequencing stages, each with its own primer strategy. This section highlights several practical issues regarding each step.

**Table 2**. **Primers used in the Swift AMPLICON™ 16S+ITS PANEL.** Primers were applied to amplify the variable regions along the 16S rRNA gene as described in the kit's protocol.

| Forward primers | Reverse primers |
|---|---|
| V1_f: GAGTTTGATCMTGGCTCAG | V4_r: CTACCAGGGTATCTAATCC |
| V3_f: CCTACGGGAGGCAGCAG | V5_r: CCGTCAATTCMTTTGAGTTT |
| V4_f: GCCAGCAGCCGCGGTAA | V8_r: GACGGGCGGTGTGTACAA |
| V6_f: ATGGCTGTCGTCAGCT | V9_r: TACCTTGTTACGACTT |
| V7_f: GYAACGAGCGCAACCC | |

**Table 3**. **Swift primer pairs applied in SMURF-based reconstruction.** Multiple amplicon types are generated in a multiplex amplification of the Swift nine primer pairs. SMURF analysis considered nine forward-reverse pairs of adjacent regions. For example, two amplicon types starting with V3_f were considered, i.e., those that end with V4_r or V5_r.

| Forward primer | Reverse primer |
|---|---|
| V1_f | V4_r |
| V3_f | V4_r |
| V3_f | V5_r |
| V4_f | V4_r |
| V4_f | V5_r |
| V6_f | V8_r |
| V6_f | V9_r |
| V7_f | V8_r |
| V7_f | V9_r |

**1. Universal LNA primers for long-read Sequencing (fixed primer set):** The initial database enrichment step uses a pre-designed, fixed set of three universal LNA primer pairs (Table 1). These primers are designed to be niche-independent and should be used without modification to amplify full-length 16S rRNA genes from pooled samples of any new environment.

**2. Multi-region primers for short-read sequencing (standard primer set):** For the second stage of profiling individual samples, we recommend using a commercially available, off-the-shelf kit that amplifies multiple non-consecutive regions of the 16S rRNA gene (e.g., the 6-region Qiagen panel or the 9-region Swift panel used in this study). This standardized approach is the most straightforward and robust. The CoSMIC software is designed to work with the output of such kits, but can also be adapted for custom primer sets.

**3. Primer subset optimization for short-read sequencing (optional for budget-constrained projects):** For most projects, we recommend using all amplicons provided by a commercial kit, as this maximizes resolution, and the marginal cost of adding primer pairs is often low. However, for very large-scale studies where cost savings are crucial, an optional optimization can be performed. As shown in Fig 2, a user can sequence a small set of representative samples (e.g., 10-20) with the full primer panel. Analyzing the resulting data can then identify a minimal subset of primer pairs (e.g., 3-4 out of the whole set from the kit) that provides sufficient resolution for that specific niche. The large-scale project can then sequence the remaining samples using only this optimized, cost-effective subset of primers.

### Generating ground truth 16S rRNA sequences in Fig 5

**Samples.** The same samples used for the Swift processing pipeline were used to generate ground truth data.

**Library preparation and sequencing.** Extracted DNA from each sample was used with the Nextera DNA Flex Library Prep (Cat. 20018704) to prepare the libraries for metagenomic sequencing. Libraries were sequenced on the Illumina NovaSeq sequencer using the following cycles 151—8—8—151, resulting in 20-80 Mbp reads per sample (S15 Fig upper panel).

**Data analysis – Generating ground truth 16S rRNA sequences.** Ground truth sequences were extracted using phyloFlash [25], which employs a multi-step approach to identify full-length 16S rRNA sequences from metagenomic data. First, phyloFlash assembles the metagenomic reads using metaSPAdes [26] to generate metagenome-assembled genomes (MAGs). Next, it aligns the metagenomic reads to the SILVA database using a low identity threshold and retaining reads that align to 16S rRNA gene sequences. These reads are then aligned to three reference sets: (a) Trusted contigs, i.e., full-length 16S rRNA gene sequences obtained by either Loop Genomics sequencing or PacBio; (b) 16S rRNA genes extracted from the metaSPAdes assembly of each sample using Barrnap [27]; and (c) The SILVA database of 16S rRNA genes, for reads that were unassigned to the two previous sources.

This alignment step allows phyloFlash to determine the composition of the bacterial community in the sample. The list of full-length 16S rRNA sequences from the trusted contigs, assembled MAGs, and SILVA database alignments served as the ground truth 16S rRNA sequences (gt16S).

**Data analysis – Improving candidate 16S rRNA sequences by MarkerMAG.** To improve the ground truth sequences for the mock mixture further, MarkerMAG [28] was applied, which uses a set of marker genes to identify and validate the presence of specific taxa in metagenomic data. The *matam_16s* script in MarkerMAG, which is based on the SILVA database, was utilized to generate a list of 16S rRNA marker sequences for the taxa present in our sample. Next, the marker sequences were linked to contigs assembled by metaSPAdes using the MarkerMAG/link-16s.py script in MarkerMAG. This step allowed us to validate the marker list by confirming that the marker sequences were linked to the assembled contigs.

To evaluate alignment identity and the erroneous frequency, the validated marker sequences of the mock mixture were compared against the output of CoSMIC.

## Calculating alignment identity, explained frequency, erroneous 16S rRNA sequences, and ambiguity in Fig 5

The output of SMURF is a set of groups, where each group includes 16S rRNA genes whose sequence is identical over the amplified regions yet varies over the rest of the gene. An analogous definition of a group for standard analysis was created based on the SILVA database, in which we extracted all relevant 16S rRNA gene sequences that share the same taxonomy set by the QIIME2 pipeline. For example, if QIIME2 provides a genus-level taxonomy for a certain ASV or OTU, its "group" comprises all SILVA sequences of this genus. This definition is equivalent to SMURF's groups as all 16S rRNA sequences sharing the same taxonomy are indistinguishable based on the reads provided.

Groups of QIIME2 or CoSMIC were used for the comparisons below:

**Ambiguity (Fig 5D):** Ambiguity is the number of full-length sequences in a group. Fig 5D displays a histogram of group sizes for both CoSMIC and QIIME2-based analyses.

The other measures in Fig 5 required processing the groups. The sequences in each group were clustered using CD-HIT, and the most frequent sequence was selected as the group's representative.

**Alignment identity (Fig 5A):** Each representative sequence was aligned to its corresponding gt16S list using Blast, and the gt16S sequence with the highest alignment was selected. The unweighted average of Blast's identity score served as the alignment identity.

**Explained frequency (Fig 5B):** Overall relative frequencies of groups whose representative 16S rRNA gene sequences aligned to gt16S for a given alignment score.

**Erroneous 16S rRNA gene sequences (Fig 5C):** Percentage of 16S rRNA sequences that did not appear in gt16S out of the total number of sequences detected by each method.

## The CoSMIC software package

The software package (https://github.com/NoamShental/CoSMIC) provides a comprehensive set of scripts facilitating all steps of CoSMIC. The package includes scripts augmenting an existing database by full-length 16S rRNA sequences obtained from long-read sequencing (e.g., PacBio). It then provides code that formats the augmented database in a SMURF-like format per the specific primers' set to be used in the short-read sequencing experiments. The package includes scripts for denoising and demultiplexing paired-end Illumina short reads using the DADA2 algorithm and running SMURF. The package supports SMURF profiling based on any subset of the original set of primers.

**Hardware requirements.** The CoSMIC package was optimized for efficiency, requiring a 20-core machine with 16GB of RAM per core, a specification within reach for most academic settings. With adjustments, our processes can run on standard PCs, albeit more slowly, making our pipeline accessible and practical for a wide range of research environments without the need for exceptional computational resources. Documentation for code and processing requirements is provided at the GitHub repository.

## Results

### The outcome of microbial profiling highly depends on amplified regions

To demonstrate the effect of preselecting a single variable region (formerly shown, e.g., by [5,6]), we performed 16S rRNA profiling of leaves, roots, and their surrounding soil from seven different plants (see Methods). Profiling was conducted using the commercial Qiagen QIAseq 16S/ITS Region Panel [29], which amplifies six regions along the gene (Fig 1A), resulting in 200K-600K reads per sample (S1 Fig). We used Qiagen's CLC Genomics Microbiome package, which applies a closed-reference Operational Taxonomic Units (OTU) picking strategy using a similarity threshold of 97% (see Methods). The package independently analyzes each region by assigning taxonomy to OTUs detected in a region and outputs the taxonomies detected across all regions.

Fig 1B illustrates the number of OTUs detected in leaf, soil, and root samples of *Aster poliothamnus* Diels (Asteraceae) using each of the six amplified regions. For example, the same root sample results in 600-1100 OTUs, depending on the

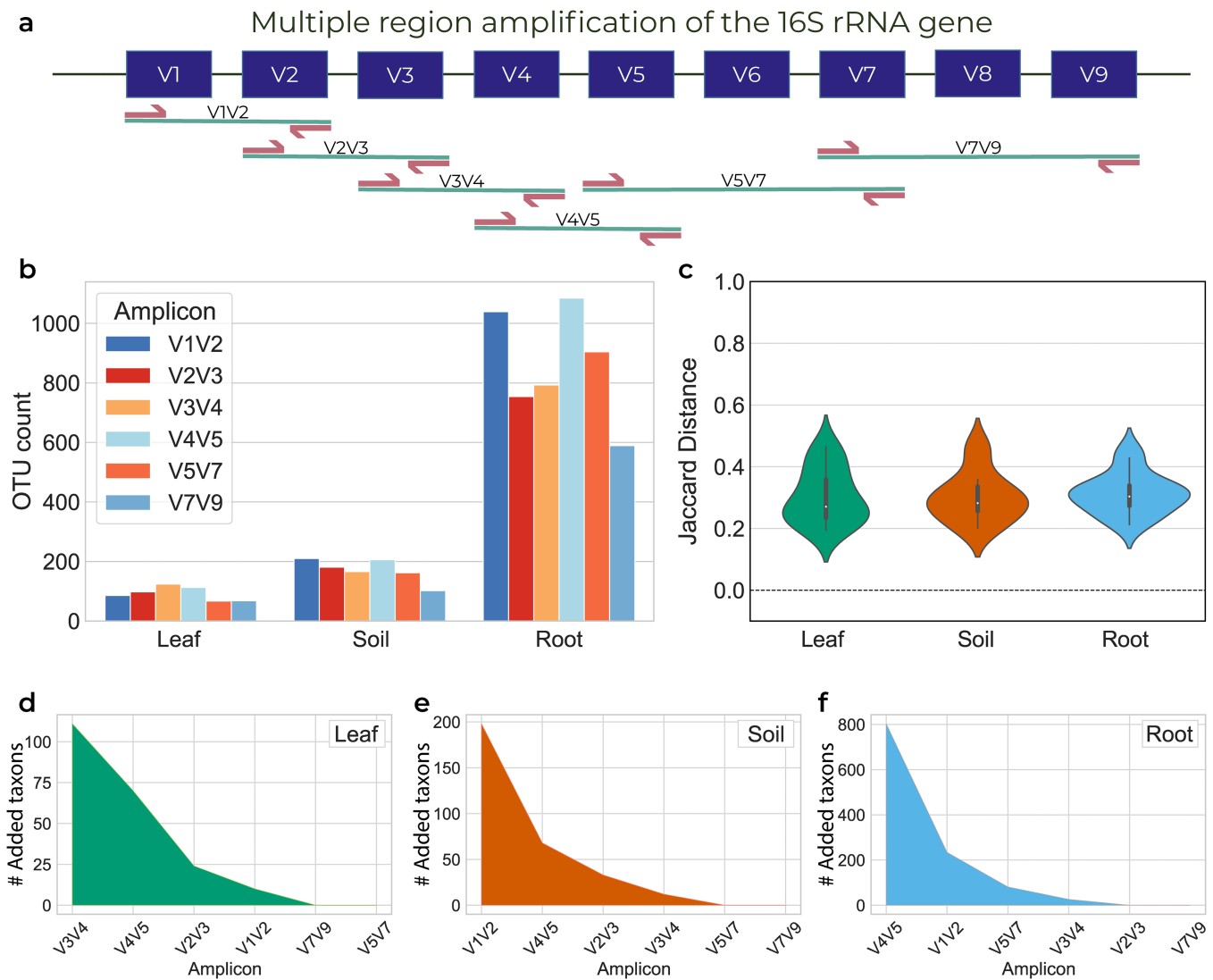

**Fig 1. Microbial profiling of *Aster poliothamnus* samples depends on the applied amplicon.** Leaf, soil, and root samples of *A. poliothamnus* were profiled by the Qiagen kit that provides six amplicons along the 16S rRNA gene, and their reads were subsequently analyzed using the CLC Genomics platform. **a.** The six amplicons of the Qiagen kit, each amplifying two or three adjacent variable regions. Taken together, these amplicons cover all variable regions. **b.** The number of OTUs detected for soil, leaf, and root samples of *A. poliothamnus* for each of the six amplicons. **c.** Distribution of Jaccard distances among pairs of amplicons. In each pair, e.g., V1V2 vs. V3V4, a Jaccard distance is calculated between the lists of taxa detected by CLC Genomics in each region. **d-f.** Decay graph of the number of unique taxa identified by the amplicons analyzed. Amplicons are ordered by the number of unique taxa a region contributes to the former set of regions. D-Leaf, E-Soil, and F-Root.

amplified region (Fig 1B, right). Moreover, there is a low overlap in assigned taxonomies detected by different regions, as evident by the beta diversity among all pairs of amplified regions (Fig 1C). Beta diversity was calculated by a Jaccard distance and is higher than zero for any pair of amplified regions (median Jaccard distances of 0.27, 0.30 and 0.28 for leaf, root, and soil, respectively).

To evaluate the contribution of each amplicon to the analysis of a given sample type, we identified the amplicon having the largest number of unique OTUs and sequentially added regions according to their additional unique OTUs with respect to regions selected thus far (Fig 1D–1F). The "top" region varied across samples, e.g., the most "informative"

regions for leaf, soil, and root in the case of *A. poliothamnus* were V3V4, V1V2, and V4V5, respectively. In addition, it required four regions to exhaust the number of unique OTUs. Yet, the identity of these regions varied among leaf, soil, and root samples, thus highlighting the inherent biases in region selection. Comparable observations were made when profiling the other plants - *Brandisia hancei* Hook, *Lepidium apetalum* Willd, *Lathyrus sativus* L, *Angelica hirsutiflora* Tang S.Liu, C.Y.Chao & T.I.Chuang, and *Tetraena mongolica* Maxim (S2–S6 Figs), where the most "informative" ampli-con was inconsistent both across plants for a given niche (e.g., leaves across all plants), and over niches for each plant (S7–S9 Figs).

## Combining results from several variable regions partially overcomes problems in profiling

We previously suggested employing the Short MUlti-Region Framework (SMURF) approach [10] to avoid issues related to pre-selecting a single region. In SMURF, any combination of regions is amplified, and their sequencing results are computationally combined, resulting in a single coherent profiling solution. SMURF relies on a database of full-length 16S rRNA sequences to solve an optimization problem, seeking the gene sequences and their relative frequencies that have given rise to the set of reads from the relevant regions. The output of SMURF is a set of "groups" and their rel-ative frequencies, where each group contains full-length 16S rRNA genes having an identical sequence over the amplified regions (i.e., these sequences differ across the rest of the 16S rRNA gene, e.g., in regions that were not part of the primer set). For example, suppose V1V2 and V7V9 regions are being amplified. In that case, full-length 16S rRNA genes having the same sequence over these two regions are indistinguishable and, if detected by SMURF, would be part of the same group. Apart from circumventing the issue of selecting a single region with its associated inherent complications, SMURF provides higher profiling resolution. The larger the number of regions, the smaller the size of a characteristic group, typi-cally allowing the identification of a single full-length gene sequence that unambiguously appears in the sample.

We applied SMURF to analyze all eight plants across leaf, soil, and root samples. Every sample underwent analysis through all 63 region combinations of the amplified six regions. For example, an analysis was performed using solely V1V2, by V1V2 combined with V3V4, followed by V1V2 combined with V3V4 and V7V9, and so forth.

Our assessment of each region's contribution was based on three parameters: (a) ambiguity, i.e., the average group size detected by SMURF. The lower the ambiguity, the higher the resolution as could be expected from the added regions; (b) relative detection, representing the fraction of identified 16S rRNA sequences out of those identified when all six regions are used (results based on all six regions are the best possible in this case; hence, all subsets are compared to them); and (c) Pearson correlation between the frequencies of 16S rRNA sequences detected by a set of regions vs. those detected by the six-region combination.

The average performance across all three parameters improved when more regions were incorporated into the SMURF analysis. The average ambiguity across all 24 samples was 20,000 sequences when using a single region. Yet, by adding only two more regions to the analysis, ambiguity was reduced to less than 2,000 (Fig 2A). Therefore, based on the reads from each additional region, full-length 16S rRNA sequences that were once part of a group are being excluded. Sec-ondly, the capacity to accurately detect a substantial proportion of the mutual species increased linearly (Fig 2B). Since sequences belonging to the same group differ by un-sequenced regions, including each additional region removed sequences that were inaccurately declared by the smaller number of regions. Thirdly, as more amplicons are added, the correlation improves significantly, with many samples achieving a correlation coefficient (r) higher than 0.9 by adding just two or three regions (Fig 2C). Fig 2A–2C displays a significant variance in performance for a given number of regions, depending on their identity. Assuming that costs scale with the number of amplified regions, the optimal selection of primers may be crucial. Such selection can be tailored to suit the specific aims and budget. One can identify an optimal selection by visualizing mutual taxonomies and Pearson correlation along the horizontal and vertical axes, respectively, and ambiguity is designated by the marker size (Fig 2D–2E). For example, combining three primer pairs, V1V2, V2V3, and V4V5, yields better results in soil samples than some sets of four primers (Fig 2D, red cross vs. all other pink circles).

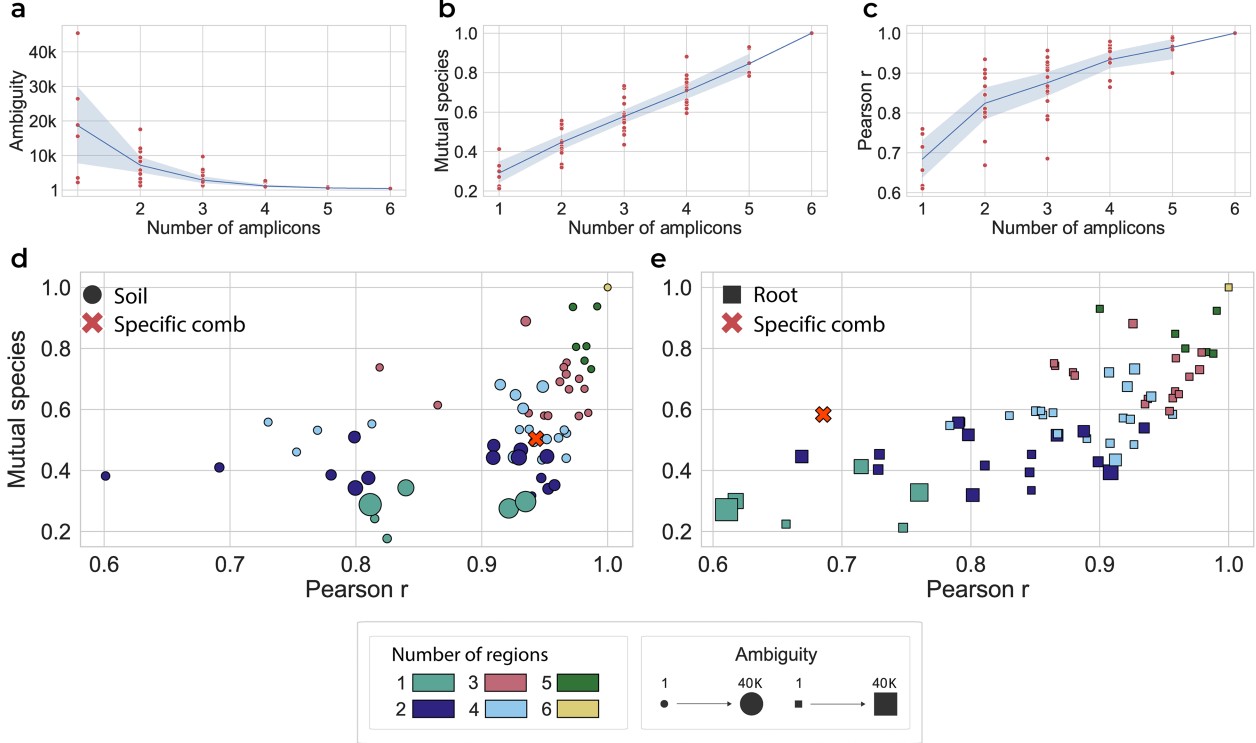

**Fig 2. Evaluating the performance of sets of primer pairs across niches.** Twenty-four samples corresponding to the leaf, soil, and root of eight plants were profiled by SMURF, using all 63 combinations of the six amplicons. **a.** Ambiguity, i.e., the number of 16S rRNA gene sequences per group as detected by SMURF, as a function of the number of amplicons. Each dot corresponds to the average ambiguity in one of the 63 distinct cases. The blue line and gray background correspond to the average and standard deviation of the ambiguity across combinations of a specific number of regions. **b.** We consider SMURF's solution based on all six regions the gold standard and display a fraction of species identified by subsets of the six regions. Shown is the fraction of species whose taxonomy matches the six-region solution. **c.** Pearson correlation between profiling based on all six regions and each set of regions. **d.** A comparison of amplicon sets for soil samples. Each dot corresponds to a specific region combination averaged across 24 samples, where the marker size scales with the ambiguity and axes correspond to Pearson correlation (horizontal) and mutual species (vertical). The marker's color indicates the number of amplicons used for SMURF analysis. Hence, a single marker corresponds to six regions (yellow), and six dark green dots correspond to the different groups of five regions. The marker "x" corresponds to a specific set of 3 regions comprising V1V2, V2V3, and V4V5. **e.** The same for root samples.

However, the same combination performs poorly for root (Fig 2E, red cross) and leaf (S10 Fig, red cross). Performance across the three measures in Fig 2 is highly consistent within a given niche. For example, the best primer pairs, chosen based on average ambiguity across soil samples, also have the lowest ambiguity variance, indicating consistent results across soil samples (S18 and S19 Figs for soil, root, and leaf).

### Enriching the database using long-read sequencing

Although SMURF provides improved profiling, its results depend on the comprehensiveness of the available full-length 16S rRNA sequence databases. However, these databases are highly enriched for human-associated bacteria, while other habitats are often poorly covered.

To bridge this gap and harness the advantages of SMURF across habitats, we propose a simple preceding step in a microbiome sampling project designated to provide niche-specific full-length 16S rRNA sequences. The procedure entails a PCR step, utilizing a generic pre-designed multiplexed set of locked nucleic acid (LNA) primers aimed at effectively amplifying the whole 16S rRNA gene (see Methods and S17 Fig). LNA primers [15,30] are shorter than regular primers (10-13nt instead of ~20nt), thus allowing an increased universality compared to regular primers while maintaining primer

specificity and high Tm (see Methods). We, therefore, propose to pool samples from the relevant niche, apply PCR amplification using the pre-designed LNA primers, and sequence by long-reads. To account for potential chimeras, the resulting reads undergo several quality assurance steps and are clustered at 99% identity to remove redundancy. A representative sequence from each cluster is aligned to the SILVA NR99 database [11], and sequences whose alignment to SILVA is lower than 99% are considered novel (for a detailed description of the abovementioned steps, see Methods). We refer to the database containing SILVA and the novel sequences as "augmented SILVA".

These pools were sequenced using PacBio's HiFi circular consensus sequencing (CCS), which generates reads with over 99.5% per-base accuracy by sequencing the same molecule multiple times [18]. The resulting reads went through a rigorous quality control process that included chimera removal (see Methods, "Filtering potential chimeras") and strict quality filtering (Q-score >40). To address any remaining sequencing errors and eliminate redundancy, the filtered sequences were then clustered at 99% identity. This clustering step prevents single-base errors or minor variants from artificially increasing the count of new sequences, as they are combined into a single representative. This entire process resulted in approximately 200K high-quality, full-length 16S rRNA sequences (S11 Fig) with an average length of 1600 bp (S12 Fig).

To illustrate this idea and estimate the number of novel sequences that can be detected, we used this approach over pools of samples, each from a different niche: (a) sponges of the species *Spongia officinalis* collected from the Mediterranean coast of France; (b) *S. officinalis* collected from the Mediterranean coast of Israel; (c) plants from the Arava desert in Israel (climate type, BWh defined as hot desert climate with annual rainfall <100mm); and (d) plants from the temperate lowland in Israel (climate type, Csa defined as a Mediterranean climate with hot, dry summers and mild, wet winters). These pools were sequenced using PacBio long reads amplicon sequencing[18], resulting in  200K high-quality, full-length 16S rRNA sequences (S11 Fig) of average length 1600 bp (S12 Fig). The correlation between sequence length and the number of reads was negligible, i.e., Pearson r = -0.012, p = 1.8e-5, thus excluding the possibility of amplification-related length biases (see S20 Fig).

Fig 3A presents the percentage of novel sequences as a function of the identity threshold when aligning to SILVA. Sponges seem to have significantly higher percentages of novel sequences [31], irrespective of identity threshold. Using the SILVA definition of a novel full-length 16S rRNA sequence, i.e., a threshold of 99% identity (vertical dashed line in Fig 3A), more than 95% of detected sequences were novel for samples of sponges and about 85% for samples taken from plants. These novel sequences include many whose relative abundance in the pool is high, especially in sponges. Fig 3B shows histograms of the number of PacBio reads across detected 16S rRNA sequences for the pools of plants and sponges (left and right columns, respectively) when split into known SILVA sequences and novel sequences (upper and lower rows, respectively). For sponges, 98% of PacBio sequences are novel, compared to 90% in plants. Moreover, 17 novel sponge sequences appear in more than 1000 reads, potentially corresponding to abundant sequences, compared to zero such sequences in plants (analogously, 214 and 25 sequences have more than 100 reads for sponges and plants, respectively). Hence, LNA amplification and sequencing detect novel sequences, some of potentially significant frequency, especially in the less-explored sponge niche. When subsequently performing short-read sequencing over individual sponge samples and performing SMURF using the augmented SILVA, more than 93% of the relative frequency in each sample was attributed to novel sequences (Fig 3C, two left columns). Therefore, standard SMURF is inadequate for these sponge samples. In contrast, on average, 90% of our plant samples were attributed to known SILVA sequences (Fig 3C, two right columns).

Another way to evaluate the significance of novel sequences is to calculate the percentage of short reads assigned to each database, thus estimating its "utility" (Fig 3D). More than 82% of short reads in each sponge sample align to augmented SILVA compared to 58% that align to SILVA (Fig 3D upper panels). We note that each short read may align to SILVA and novel sequences, yet the additional 24% of the reads aligned to only novel sequences "drive" SMURF's optimization to reject SILVA sequences and select novel sequences as part of the profiling solution. The case for plant samples differs as more than 80% of the reads are aligned to SILVA, and novel sequences contribute only an additional 8%

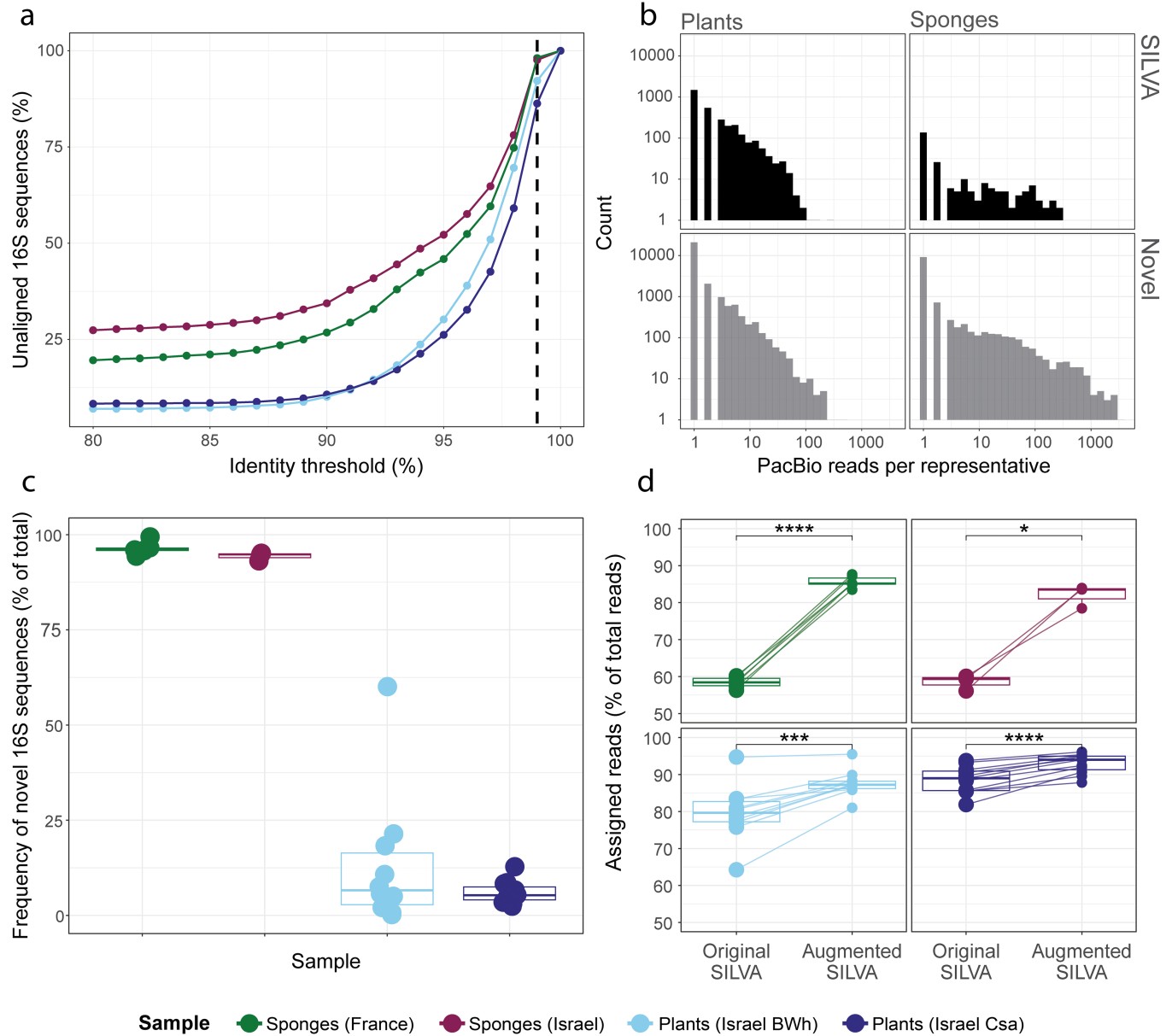

**Fig 3**. **Enriching the database with novel full-length 16S rRNA sequences.** We amplified the full-length 16S rRNA gene using pre-designed LNA primers and sequenced the resulting amplicons by PacBio long-read sequencing. Four samples were processed, each containing a pool of samples from a different habitat, i.e., plants from the Arava desert in Israel, characterized by a climate type BWh (defined as hot desert climate with annual rainfall <100mm), according to Köppen climate classification; plants from the lowland region in Israel, characterized by a climate type Csa (defined as a Mediterranean climate with hot, dry summers and mild, wet winters); and pools of samples from the marine sponge *S. officinalis* collected in the Mediterranean Sea along the coast of either France or Israel. **a.** All non-redundant PacBio sequences (Plants BWh - 14852, Plants Csa - 14589, Sponges-France - 6507, Sponges-Israel - 5468) were aligned to SILVA using the CD-HIT algorithm [20]. The percentage of sequences that did not appear in SILVA is shown as a function of the identity threshold. The vertical dashed line corresponds to SILVA's standard threshold for calling novel full-length 16S rRNA sequences (SILVA NR99). Colors indicate different biomes sampled. **b.** Histograms of read counts per 16S rRNA sequence for the plants' pool (left panels) and the sponges' pool (right panels). The upper and lower panels show histograms of SILVA and novel sequences, respectively. An identity threshold of 99% was used to call novel sequences with respect to SILVA. **c.** Individual samples from the abovementioned pools were sequenced via short reads and analyzed by SMURF using the augmented database to determine their microbial composition (Sponges-France – 6 samples, Sponges-Israel – 3 samples, Plants (BWh) – 10 samples, Plants (Csa) – 10 samples). The total relative frequency assigned by SMURF to novel 16S rRNA sequences per sample (dots) and across types (columns) is shown. **d.** The fraction of reads aligned to either SILVA or augmented SILVA for each sample depicted in C. Each panel corresponds to a different sample type. To evaluate the statistical significance, a paired t-test was performed for each sample type; * P<0.05, *** P<0.001, **** P<0.0001.

and 4% for BWh and Csa, respectively (Fig 3D lower panels). Consequently, SMURF's solution attributes most of the relative abundance to SILVA sequences.

Another observation is that 91% and 97% of the novel sequences for sponges and plants, respectively, appear in less than 10 PacBio reads (Fig 3B). Although rarefaction curves plotted over these pools do not reach a plateau for either sponges or plants (S13 Fig), indicating extensive rare biosphere diversity, the captured sequences account for >90% of community abundance in individual samples. Deeper sequencing would have identified additional rare taxa (<10 reads), which probably would have closed the remaining gap in unaligned reads.

## The CoSMIC approach

When considering the problems and solutions presented in the former sections, we propose guidelines for large-scale microbiome studies that would provide high-resolution profiling in a new or known niche while accommodating different budget tiers (Fig 4). We recommend treating each research project as a new ecological niche (orange, right-hand side in the flowchart) and enrich current databases with relevant full-length 16S rRNA gene sequences, i.e., pool multiple samples of different origins of the ecological niche, apply LNA amplification and perform long reads sequencing. Next, using the newly enriched database, we propose to sequence a representative set of samples using the most extensive set of primers available (e.g., all nine Swift pairs or all six Qiagen pairs), apply SMURF, and select the subset of primer pairs that provide sufficient resolution per niche (or select all primer pairs in case costs do not scale considerably with the number of primers). Once these steps are finalized, the project transitions to a "familiar niche" case (purple, left-hand side), which performs routine amplification of the optimal primer set followed by short reads sequencing, subsequently analyzed by augmented-database SMURF.

## Experimental evaluation of CoSMIC

To evaluate the advantages of CoSMIC, we sampled *Thymelaea hirsuta* (L.) Endl, *Solanum lycopersicum* L., and *Triticum aestivum* L. at their leaves, roots, and surrounding soil. In addition, we created a mock community of 12 known species. To construct our ground truth, we performed standard shotgun metagenomic sequencing of each of the ten samples and used phyloFlash [25] to extract 16S rRNA sequences and their relative frequencies in each sample (see Methods). We refer to these sequences as ground truth (gt16S), and their relative frequencies are called ground truth frequencies. An optimal performance corresponds to detecting the exact full-length sequences of gt16S while adding no other sequence. Note that each species in the community may contain several different 16S rRNA sequences, thus the task is to detect all these sequences (To further support our performance, we also successfully validated our pipeline using a commercially available standard mock community, achieving complete accuracy; see Methods and S21 Fig).

As the first step in CoSMIC, we pooled DNA from all ten samples and used our LNA primers to amplify the full-length 16S rRNA gene. We sequenced the amplicons using synthetic long reads technology (Loop Genomics) to achieve full-length 16S rRNA gene sequences. Novel sequences, with respect to the SILVA NR99 standard, were added to SILVA to create our reference database.

Each sample was then sequenced using Illumina short reads using the Swift pipeline, which amplifies several regions along the 16S rRNA gene (see Methods section for detailed information about obtained reads for Loop, Swift, and shotgun metagenomics). To apply SMURF, we used our reference database while considering nine amplicons spanning the 16S rRNA gene based on the above mentioned primers.

For comparison, we provide a standard analysis provided by Swift utilizing the QIIME2 [22] package to generate tables of operational taxonomic units (OTUs) or amplicon sequence variants (ASVs) utilizing DADA2 [23], each based on either all amplified regions or only the widely used V3V4 amplicon.

Comparison to the ground truth of each sample was based on four criteria: (a) *Alignment identity* - the average alignment score between detected 16S rRNA sequences and gt16S; (b) *Explained frequency* - the sum of ground truth

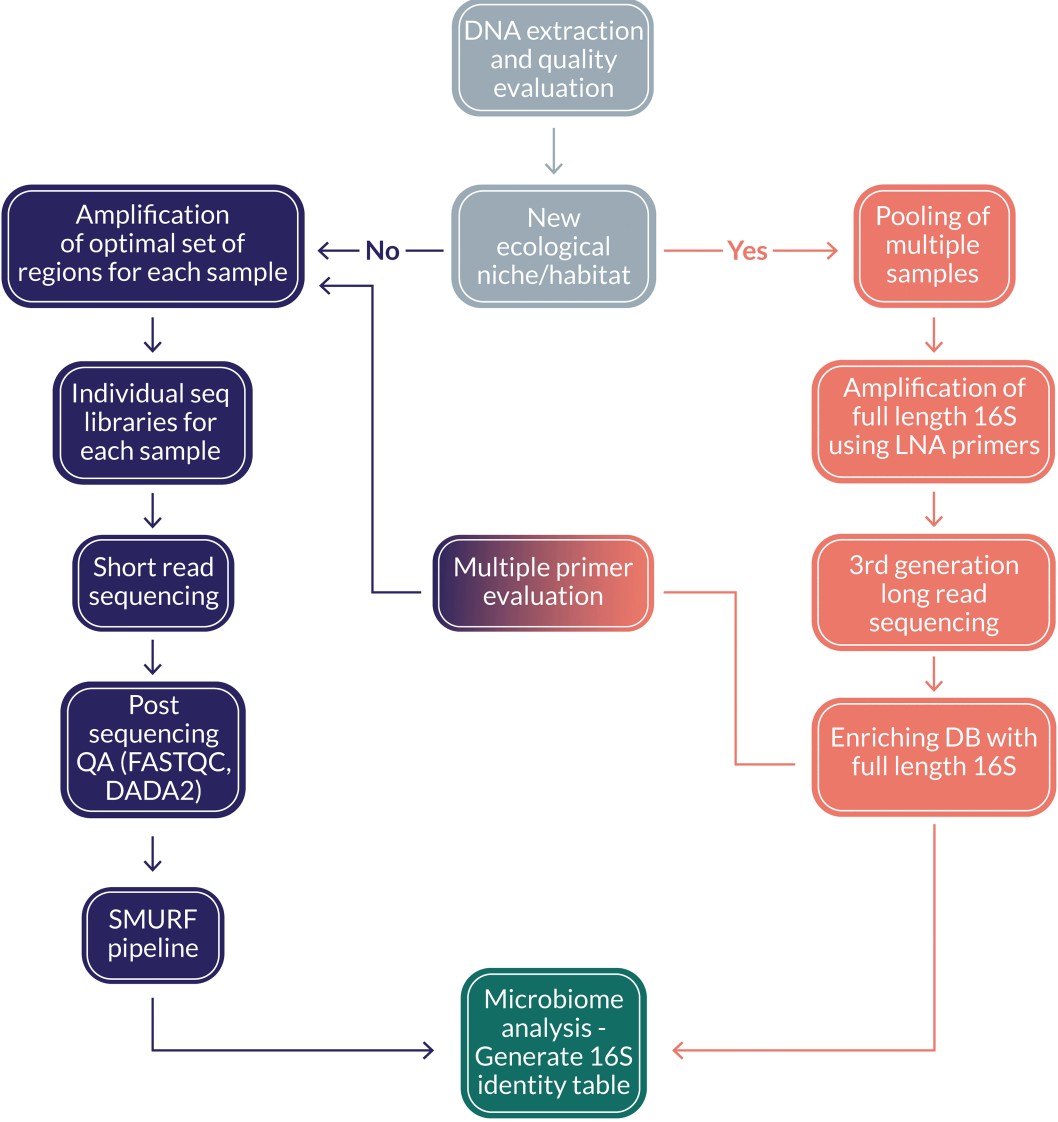

**Fig 4**. **The suggested CoSMIC flowchart.** Gray - Perform standard DNA extraction from environmental samples. Orange - Establish a niche-relevant full-length 16S rRNA gene database by LNA-based amplification and long-read sequencing. Orange-purple - Test multiple primer sets to find an optimal primer pair combination. Purple - Perform high throughput profiling in a large-scale project. Samples are amplified using the optimal primer combination and analyzed by SMURF based on the augmented database.

relative frequencies for gt16S that were also detected by each method; (c) *Erroneous 16S rRNA genes sequences* - the number of detected sequences that were not present in gt16S; and (d) *Ambiguity* - the number of sequences in the reference database that match detected sequences by each method (for details regarding a-d see Methods).

CoSMIC outperforms traditional methods on all parameters (Fig 5).

*Alignment identity* - Mean and overall alignment identities are higher for CoSMIC across all samples (Fig 5A). The distribution varies according to the tissue sampled, with mock, root, and soil consistently producing better scores than leaf tissue. This is likely related to the low number of bacteria detected due to chloroplasts occupying most reads. While the difference between analysis methods is evident across all tissues, the most striking example is the mock community

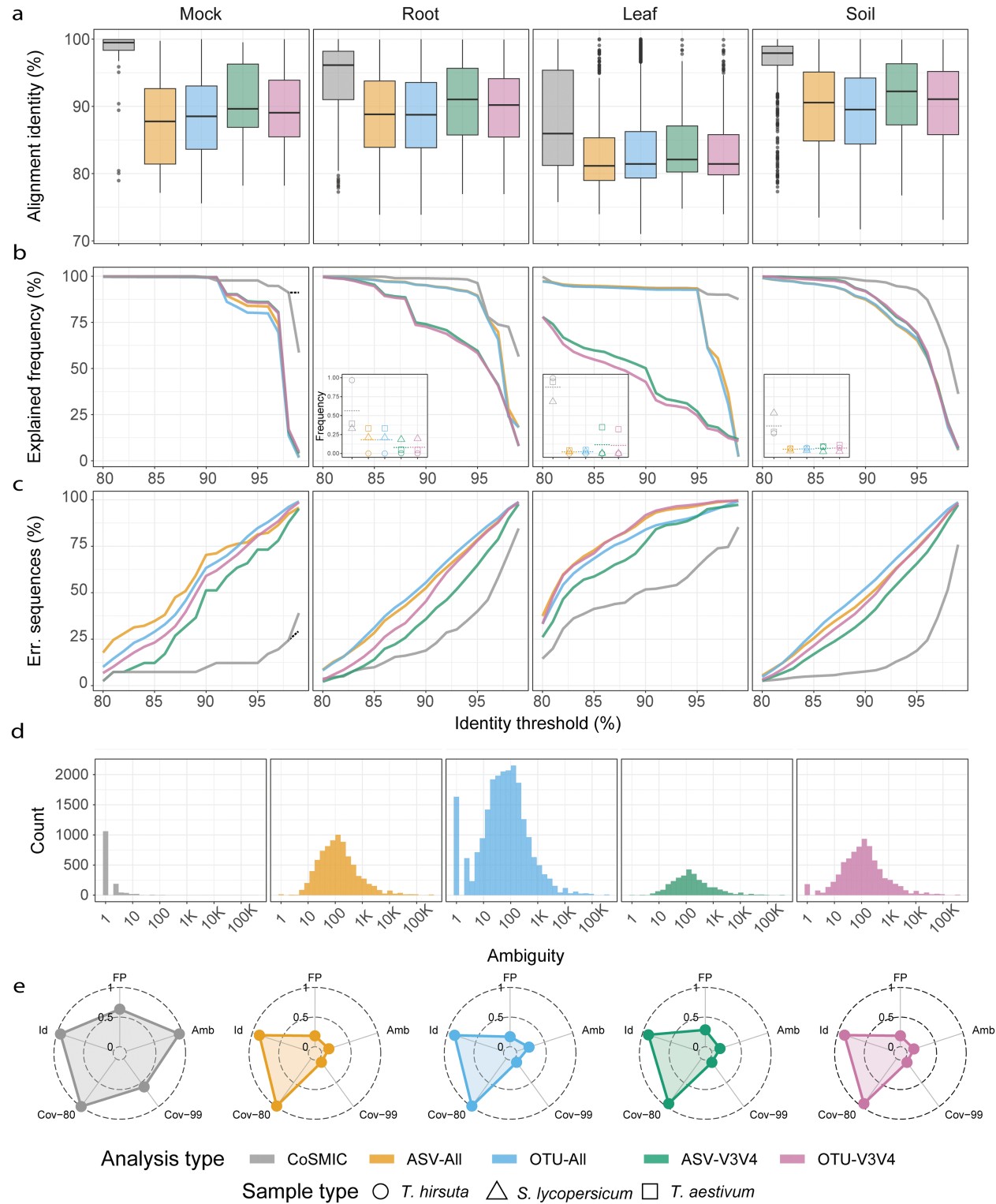

**Fig 5. Experimental evaluation of standard methods vs. CoSMIC.** Ten samples were sequenced - leaf, soil, and root samples of *T. hirsuta*, *S. lycopersicum*, and *T. aestivum*, and a mock mixture of 12 known species (see Methods). Each sample underwent standard shotgun metagenomic sequencing, from which we detected its full-length 16S rRNA sequences to be considered ground truth. In addition, as part of CoSMIC, we pooled extracted DNA from all ten samples, amplified by LNA primers, and sequenced via synthetic long reads (Loop Genomics) to create full-length 16S rRNA sequences that were added to the SILVA database. Standard amplicon sequencing was performed by a commercially available kit by Swift, which uses

primers spanning the 16S rRNA gene to amplify several variable regions (see Methods). Analysis was then performed by CoSMIC using nine primer pairs and by a standard QIIME2 pipeline. The latter used either nine primer pairs (All) or the V3V4 region only (V3V4) and was performed using OTUs or by ASV calling. Hence, four QIIME2-based results are shown – OTU-All, OTU-V3V4, ASV-All, and ASV-V3V4. **a.** Alignment identity, i.e., the average alignment score between detected 16S rRNA sequences and ground truth 16S rRNA genes (gt16S) detected by shotgun metagenomics. Boxplots present the distribution of alignment identities for each tissue and analysis method. Leaf, soil, and root panels correspond to an average of over three samples (one from each plant). A Mann-Whitney U test was performed for all comparisons, which resulted in very low p-values, with the highest being 0.0064. **b.** The total frequency of correctly detected gt16S by each method as a function identity threshold. The dotted horizontal line in the left panel (mock mixture) marks the results if we improve the ground truth sequences by MarkerMag. Insets present the total frequency achieved per sample at a cutoff of 99%; the dotted horizontal line is their average. The y-axis, shown for the leftmost inset, is relevant for all insets. **c.** Percentage of erroneous 16S rRNA sequences, i.e., those identified by an analysis method yet not detected by shotgun metagenomics. The percentage is a function of the identity threshold, where alignments below the threshold are considered erroneous. The dotted line marks the MarkerMag-based improvement for the mock community using CoSMIC for up to a 99% identity threshold. **d.** Histogram of the number of taxa matching an identification by each analysis type. **e.** Radar plot combining all samples over several criteria (normalized on a 0-1 scale, where large values correspond to better performance): Id, mean alignment identity for each analysis method as described in A; Cov-80, mean total ground truth frequency at 80% identity threshold as described in B; Cov-99, the same for 99% identity threshold; FP, One minus the mean false positive rate as described in C; Amb, mean ambiguity as described in d.

scores. Swift-based analysis, based on all regions or only on V3V4, had a median alignment identity of about 90% even for this simple, well-defined mixture, compared to an almost 100% alignment identity for CoSMIC.

*Explained frequency* - Significant improvements are also apparent in the accuracy of community composition. Fig 5B presents the total explained frequency as a function of the identity threshold averaged per sample type. Using CoSMIC, more sequences are accurately reported, accounting for a higher proportion of the ground-truth sample. The difference between the methods is particularly evident when only high-quality alignments are considered, resulting in almost zero frequency explained by traditional methods.

In three of the four cases, such reduction also seems to affect CoSMIC, yet to a lesser degree. To understand this phenomenon, we considered the mock mixture whose species composition is known. The significant drop in explained frequency occurs at an identity threshold of 98.4% and mainly originates from a mismatch between an *E. coli* sequence in gt16S and the detected CoSMIC sequence. Aligning Illumina short reads of the Swift pipeline to both sequences showed a perfect match to the detected CoSMIC sequence, while the gt16S *E. coli* sequence contained 12 SNPs relative to the reads. Therefore, the actual value of CoSMIC's explained frequency at identity thresholds higher than 99% is at least 88%. This indicates that phyloFlash-derived 16S rRNA sequences may not be entirely accurate, demonstrating the inherent problems in achieving highly accurate assembly of 16S rRNA sequences from shotgun metagenomics. This problem is not restricted to phyloFlash and also characterizes other state-of-the-art assemblers [32,33]. To alleviate this problem yet provide an estimate solely based on shotgun results, we used MarkerMAG [28] to improve the gt16S sequences (see Methods), yielding an explained frequency of 89% for identity thresholds higher than 99% (Fig 5B left panel, grey dashed line). This also indicates that most of the improvement in gt16S is attributed to the errors in the abovementioned *E. coli* sequence.

*Erroneous 16S rRNA sequences* - The percentage of erroneous sequences, i.e., those reported yet not present in the ground truth sample, is significantly lower for CoSMIC than the other methods (Fig 5C). Moreover, in three of the four cases, the number of erroneous sequences using standard methods grows linearly with the identity threshold, while CoSMIC displays an increase starting from a threshold of 95%. To understand this phenomenon in the case of the mock mixture, we used the MarkerMAG-improved sequences as ground truth, yielding a reduction of 9.5% (Fig 5C left panel, grey dashed line). Although the percentage of erroneous sequences at a high identity threshold is still significant, we note that since MarkerMAG-based sequences explain 89% of the mock mixture sequences at an identity threshold of 99%, the erroneous sequences correspond to only 11% of the population. Moreover, the absolute number of these erroneous sequences is merely 12 (S14 Fig). In addition, these errors may also be another manifestation of the limitations of shotgun-based ground truth. Although more than 60 million reads were allocated to shotgun sequencing of the mock

mixture (S15 Fig), only  88,000 were assigned to 16S rRNA sequences. Hence, the ground truth may be missing some sequences, and the actual erroneous sequences at 99% identity may be lower than 12.

*Ambiguity* - Fig 5D presents a histogram of the ambiguity across all 16S rRNA sequences detected in the ten samples. Most OTUs or ASVs detected by Swift correspond to hundreds or thousands of the database's 16S rRNA sequences, while CoSMIC detects a single sequence, yielding a more precise, actionable result.

Fig 5E combines all criteria using radar plots for each method, clearly showing CoSMIC's improved performance, especially in addressing ambiguity (Amb) and explained frequency at high alignment identity scores (Cov-99). CoSMIC outperforms standard pipelines regardless of methodology (OTU or ASV) and whether multi-region or single-region analysis is used. While standard methods can identify species (Id) at low sequence resolution (Cov-80), they often result in many erroneous species (FP) not present in the sample and have very high ambiguity (Amb). As expected, adding more regions to the standard analysis does little to improve results since these methods do not combine inputs from the different regions.

## Discussion

The diverse applications of microbiome studies have underscored the need for robust and standardized methodologies [34]. However, primer selection and arbitrary analysis decisions made by standard methods lead to lower profiling resolution (Fig 5A, 5D) and fall short due to the omission of species (Fig 5B) or the inclusion of species that are not present in the sample (Fig 5C).

To address these limitations, we formerly proposed applying sets of primers from different variable regions and computationally combining their results via the SMURF approach [10]. To do so, SMURF requires a comprehensive database of full-length 16S rRNA sequences. Yet, such comprehensiveness holds mainly for human-associated samples, and SMURF's performance is sub-optimal for other niches (e.g., Fig 3C–3D). CoSMIC aims to bridge this gap in a cost-effective way by introducing a preliminary step of proactively enriching the database with niche-relevant full-length 16S rRNA sequences. We pool samples from the novel niche and apply amplification by our generic LNA-based (almost) universal primers followed by long-read sequencing. Subsequently, one can apply short-read sequencing over each sample and apply SMURF using the augmented database to achieve high-resolution, more accurate profiling results (Figs 3C–3D and 5). Applying CoSMIC is more important in less explored environments, e.g., marine sponges, where close to 100% of the reads align to novel 16S rRNA gene sequences detected by CoSMIC (Fig 3C). Therefore, we hope CoSMIC will be adopted by researchers seeking efficient and accurate large-scale profiling of less explored niches.

To our surprise, CoSMIC also outperformed standard methods when profiling a mock community of known species. The average alignment score between 16S rRNA sequences detected by CoSMIC and the ground truth sequences was 97.2% (Fig 5A left panel), while the score of standard methods was lower by  8%. This corresponds to a  120bp difference along the gene, thus potentially leading to a lower resolution. Also observed was an overestimation of the diversity of 16S rRNA sequences over the mock mixture, evident in both ASV or OTU-based standard analysis (Fig 5C left panel). Hence, CoSMIC provides a more robust estimation of this important ecological property.

Overall costs per sample in CoSMIC (including all expenses - from DNA extraction to those related to short- and long-read sequencing) currently range between $25 to $30, depending on the number of primer pairs applied. Hence, overall costs only marginally depend on the number of primer pairs since the expensive stages are DNA extraction and other sample preparation steps, which do not depend on the number of applied primer pairs. Also, amplification is often applied in multiplex; hence, sample preparation and labor may be comparable to using a single primer pair. Moreover, large-scale projects may benefit from a 10-15% reduction in costs associated with the number of applied primer pairs. Therefore, we advise researchers to estimate the minimal subset of primer pairs that provide adequate results for their samples (e.g., Fig 2).

The straightforward approach of using our generic LNA primers and applying PacBio or Oxford Nanopore [35] long-read sequencing to individual samples rather than pools of samples is possible and would achieve equivalent results to CoSMIC. However, its costs become prohibitive for large projects. For example, the estimated commercial cost for PacBio sequencing or Oxford Nanopore long-read sequencing is about $30 per sample, yielding 10K reads per sample (prices scale almost linearly for a higher number of reads). This price tag allows about 4M short reads per sample in CoSMIC, roughly corresponding to more than 600K "long reads" when amplifying six regions by the Qiagen kit. Hence, CoSMIC provides a means of providing de facto long reads while circumventing high costs, which is especially beneficial for large-scale projects that require a large number of reads per sample (e.g., 100K) to detect lower abundance bacteria.

Researchers may also turn to shotgun metagenomics techniques to retrieve 16S rRNA genes, yet the aims of shotgun sequencing differ from mere profiling, and their costs are much higher. Moreover, state-of-the-art assemblers often fail to provide highly accurate 16S rRNA gene sequences of a sample's organisms (e.g., the *E. coli* sequence in our mock mixture of Fig 5). The gene's conserved regions across species and often its almost identical copies within the same organism hampers accurate assembly. As a result, relying solely on metagenomics-based approaches for 16S rRNA profiling may provide less accurate results than CoSMIC.

Beyond these alternative workflows, researchers may consider other methods for the initial enrichment of full-length 16S rRNA sequences. Hybridization-based enrichment methods, such as SureSelect (Agilent) and xGen Lockdown probes (IDT), have been successfully applied to enrich 16S rRNA genes from complex samples [36,37]. However, we preferred using LNA primers for several reasons: (a) their shorter length allows for nearly universal amplification while maintaining specificity through locked nucleic acid modifications (hybridization probes are two to ten times longer than the 10-13 nt of the LNA primers); (b) a PCR-based approach is considerably more cost-effective than hybridization capture; and (c) the protocol is simpler, faster, and requires only standard PCR equipment, unlike specialized hybridization workflows.

The importance of proactively enriching the database with niche-specific full-length 16S rRNA sequences has been demonstrated by several studies profiling, e.g., the honey bee gut microbiome [38], wastewater treatment systems, and anaerobic digesters [39]. The latter's authors further developed a method, AutoTax, for assigning taxonomy to these novel full-length sequences. Currently, CoSMIC does not involve any taxonomic assignment, hence we intend to apply AutoTax or similar taxonomy assignment tools (e.g., TaxAss [40]) to the output of CoSMIC to assist users.

While the CoSMIC approach represents an advancement in microbiome analysis, it requires careful quality assurance by researchers. In CoSMIC, a novel niche refers to cases where a significant fraction of the reads cannot be matched to the available database of full-length 16S rRNA sequences. Hence, researchers are expected to monitor this fraction to ensure comprehensive results and re-apply the LNA step upon encountering new communities.

Adding irrelevant or incorrect full-length 16S rRNA sequences to an ad-hoc CoSMIC database does not affect its reconstruction results. For example, thousands of 16S rRNA gene sequences that appeared as single PacBio reads and were part of the augmented database did not participate in any of CoSMIC's reconstructions. Also, chimeric reads that end up in an augmented database cannot be part of CoSMIC's reconstructions since there will not be a set of short reads that supports such an erroneous sequence. However, as performed here, we advise researchers to apply chimera-detection (e.g., intrinsic algorithms by PacBio, VSEARCH uchime_denovo [19], etc.) prior to adding sequences to the ad-hoc database to keep the database "clear" of irrelevant sequences. Moreover, when compiling a set of novel sequences to be added to a public general-purpose 16S rRNA database, or a niche-specific pool of sequences, researchers should only consider sequences detected by SMURF in at least one sample. By this, databases would only contain "validated" sequences, i.e., those that passed both chimera-detection algorithms and were detected by SMURF. This would potentially allow researchers of the same niche to directly apply the short-read part of CoSMIC and skip long-read sequencing once an adequate "coverage" of the environment has been achieved.

Comparisons between CoSMIC and other methods were based on detecting the correct bacteria yet ignoring their relative frequencies. This omission is due to the fact that frequency estimates are not robust enough in all primer-based methods, including CoSMIC. These inaccuracies mainly result from noise and biases in PCR amplification, yet they also stem from different copy numbers of the 16S rRNA genes or different sequences of the gene in the same cell. To overcome this problem and allow accurate estimates of relative frequencies, we aim to utilize unique molecular identifiers (UMIs) as part of the CoSMIC sample preparation, similar to our former quantification method [41]. UMI tagging individual template molecules prior to amplification may allow downstream computational correction of varying amplification efficiencies.

## Conclusion

In conclusion, employing CoSMIC will circumvent the problems caused by standard profiling methods, allow harmonizing results across studies from the same niche or from different niches, and potentially aid in inferring species functions and biological importance. We hope that this framework, allowing low-cost, high-resolution microbial profiling of any niche, would provide a modular and optimized solution for the evolving landscape of microbiome research.

## Acknowledgments

We thank Shira Holand for scientific illustrations and graphic design; Micha Ilan for the collections of *S. officinalis* from Israel; Thierry Pérez and Marie Grenier for the collections of *S. officinalis* from France; David Pilzer for PacBio sequencing support; and Jonathan Nutkiewitz and Elad Jacobi-Carmeli for helpful thoughts and discussions.

## Supporting information

**CoSMIC_Supp.pdf. Supporting information containing Supplementary S1–S21 Figs, S1 Table, data availability links (code and raw reads), and the No-Template Control (NTC) analysis.**
(PDF)

## Author contributions

**Conceptualization:** Maor Knafo, Shahar Rezenman, Tal Idan, Michael Elgart, Shlomi Dagan, Ziv Reich, Dagan Sade, Noam Shental.

**Data curation:** Maor Knafo, Shahar Rezenman, Tal Idan, Vered Zavaro, Dagan Sade, Noam Shental.

**Formal analysis:** Maor Knafo, Shahar Rezenman, Tal Idan, Michael Elgart, Shlomi Dagan, Dagan Sade, Noam Shental.

**Funding acquisition:** Shahar Rezenman, Ruti Kapon, Dagan Sade, Noam Shental.

**Investigation:** Maor Knafo, Shahar Rezenman, Tal Idan, Ruti Kapon, Dagan Sade, Noam Shental.

**Methodology:** Maor Knafo, Shahar Rezenman, Tal Idan, Ruti Kapon, Dagan Sade, Noam Shental.

**Project administration:** Maor Knafo, Shahar Rezenman, Dagan Sade, Noam Shental.

**Resources:** Maor Knafo, Shahar Rezenman, Shlomi Dagan, Dagan Sade, Noam Shental.

**Software:** Maor Knafo, Shahar Rezenman, Tal Idan, Shlomi Dagan, Dagan Sade, Noam Shental.

**Supervision:** Shahar Rezenman, Michael Elgart, Ziv Reich, Ruti Kapon, Dagan Sade, Noam Shental.

**Validation:** Maor Knafo, Shahar Rezenman, Ziv Reich, Dagan Sade, Noam Shental.

**Visualization:** Maor Knafo, Shahar Rezenman, Tal Idan, Vered Zavaro, Dagan Sade, Noam Shental.

**Writing – original draft:** Maor Knafo, Shahar Rezenman, Tal Idan, Ziv Reich, Dagan Sade, Noam Shental.

**Writing – review & editing:** Maor Knafo, Shahar Rezenman, Tal Idan, Michael Elgart, Shlomi Dagan, Ziv Reich, Dagan Sade, Noam Shental.

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
