## [Decision Letter · Decision Letter 0]

24 Sep 2025

PONE-D-25-22924CoSMIC - A hybrid approach for large-scale, high-resolution microbial profiling of novel nichesPLOS ONE

Dear Dr. Shental,

Thank you for submitting your manuscript to PLOS ONE. After careful consideration, we feel that it has merit but does not fully meet PLOS ONE’s publication criteria as it currently stands. Therefore, we invite you to submit a revised version of the manuscript that addresses the points raised during the review process.

The reviews of this manuscript were generally positive. There are a number of issues that need to be addressed, particularly those of reviewer #2 regarding the mock community and the lack of negative controls to assess any contamination. Both reviewers have provided detailed comments. Be sure to go over the comments carefully and respond to each issue that is noted.

We look forward to receiving your revised manuscript.

Kind regards,

Theodore Raymond Muth

Academic Editor

PLOS ONE

[Funding was provided by Sustainability and Energy Research Initiative by TI; The Yotam project and the Weizmann Institute sustainability and energy research initiative to DS; Combined Weizmann-Abroad Postdoctoral Grant for Advancing Women in Science and the Emergency Fellowships for Israeli Postdocs Affected by COVID-19 to TI. The Open University of Israel internal grant 10024 to NS.].

4. Please update your submission to use the PLOS LaTeX template. The template and more information on our requirements for LaTeX submissions can be found at http://journals.plos.org/plosone/s/latex.

Additional Editor Comments:

Reviewer #1:

Reviewer #2:

Reviewers' comments:

Reviewer's Responses to Questions

**Comments to the Author**

1. Is the manuscript technically sound, and do the data support the conclusions?

Reviewer #1: Yes

Reviewer #2: Yes

2. Has the statistical analysis been performed appropriately and rigorously?

Reviewer #1: Yes

Reviewer #2: Yes

3. Have the authors made all data underlying the findings in their manuscript fully available?

Reviewer #1: Yes

Reviewer #2: Yes

4. Is the manuscript presented in an intelligible fashion and written in standard English?

Reviewer #1: Yes

Reviewer #2: Yes

5. Review Comments to the Author

Reviewer #1: The manuscript describes COSMIC, a method that combines short and long reads to enhance 16S identification for uncharacterized microbiome niches.

The combination of short and long reads to benefit high-throughput sequencing is an interesting approach that has many implementations.

The research produces data, explores it, and presents it well.

Some minor issues need to be addressed in the current version. The most significant of them is the effect of sequencing errors on the system. This can be addressed by the two methodologies in the manuscript, PacBio and LoopSeq.

Reviewer #2: The paper "CoSMIC – A hybrid approach for large-scale, high-resolution microbial profiling of novel niches" introduces CoSMIC (Comprehensive Small Ribosomal Subunit Mapping and Identification of Communities), a methodological framework designed to enhance microbial community profiling. CoSMIC builds upon the previously developed SMURF approach. According to the authors, CoSMIC offers a practical solution for studying novel ecological niches.

The revisions made by the authors in response to the reviewers’ comments are adequate and satisfactory. However, some weaknesses remain that require further clarification and improvement.

1. The full and short titles are the same. Authors should correct this and write the short title.

2. The introduction remains too general and reiterates well-known limitations of standard 16S rRNA profiling (e.g., primer bias, short-read ambiguity) rather than emphasis the specific research problem CoSMIC addresses.

3. The final paragraph in Intruduction section presents partial results instead of clearly stating the research objectives and hypotheses. This should be restructured.

4. Additional citations are needed at the end of sentences in lines 60, 63, and 77 to support key statements.

5. I am not fully convinced by the preparation of the mock mixture. While the rationale is understandable, the analyses would be more reliable if a standardized mock community (e.g., commercially available reference mixtures) had been included.

6. Negative controls were not applied at any stage of the study. This omission makes it impossible to determine to what extent the observed microbial profiles might be influenced by contamination. Based on my own experience, even primer purification methods can significantly affect contamination levels and therefore should be considered.

7. The study does not provide an optimized, ready-to-use primer set. It appears that each user must test multiple primer combinations before identifying the most suitable ones. This lack of standardization reduces the reproducibility and usability of the approach.

8. The authors are reminded that, in accordance with the rules of biological nomenclature (ICZN/ICN), the first time a species is mentioned in the manuscript, the full species name must be provided together with the author citation. In all subsequent mentions of the same species, the genus name should be abbreviated, as required by taxonomic conventions.

9. The use of the Greengenes database is problematic, given that it has not been updated since 2013. Silva is a more appropriate choice.

10. The study does not explain why NCBI was not considered, nor does it compare results obtained with different reference databases. A comparative analysis would strengthen the conclusions and clarify which database is most suitable for CoSMIC.

11. Rarefaction curves for some samples did not reach a plateau, raising concerns about whether sequencing depth was sufficient. This calls into question the reliability of applying CoSMIC to novel and complex samples.

12. Relative abundance estimates are acknowledged as unreliable, but the discussion provides little evidence for how UMIs will solve this issue in practice. This limitation currently reduces the ecological interpretability of CoSMIC results.

13. Claims about harmonizing results across studies and niches are speculative and should be more cautiously stated, given unresolved issues with standardization, database completeness, and contamination controls.

6. PLOS authors have the option to publish the peer review history of their article (what does this mean?). If published, this will include your full peer review and any attached files.

Reviewer #1: No

Reviewer #2: No

---

## [Decision Letter · Decision Letter 1]

21 Dec 2025

CoSMIC - A hybrid approach for large-scale, high-resolution microbial profiling of novel niches

PONE-D-25-22924R1

Dear Dr. Shental,

We’re pleased to inform you that your manuscript has been judged scientifically suitable for publication and will be formally accepted for publication once it meets all outstanding technical requirements.

Kind regards,

Theodore Raymond Muth

Academic Editor

PLOS One

Additional Editor Comments (optional):

Reviewers' comments:

Reviewer's Responses to Questions

**Comments to the Author**

1. If the authors have adequately addressed your comments raised in a previous round of review and you feel that this manuscript is now acceptable for publication, you may indicate that here to bypass the “Comments to the Author” section, enter your conflict of interest statement in the “Confidential to Editor” section, and submit your "Accept" recommendation.

Reviewer #1: All comments have been addressed

Reviewer #2: All comments have been addressed

2. Is the manuscript technically sound, and do the data support the conclusions?

Reviewer #1: Yes

Reviewer #2: Yes

3. Has the statistical analysis been performed appropriately and rigorously?

Reviewer #1: Yes

Reviewer #2: Yes

4. Have the authors made all data underlying the findings in their manuscript fully available?

Reviewer #1: Yes

Reviewer #2: Yes

5. Is the manuscript presented in an intelligible fashion and written in standard English?

Reviewer #1: Yes

Reviewer #2: Yes

6. Review Comments to the Author

Reviewer #1: The revised version of the manuscript is now acceptable for publication. The authors have addressed all the issues raised in the review of the previous version. No further issues.

Reviewer #2: Dear Authors,

I am satisfied with your responses to my comments and therefore recommend accepting the article for publication in PLOS ONE.

7. PLOS authors have the option to publish the peer review history of their article (what does this mean?). If published, this will include your full peer review and any attached files.

Reviewer #1: No

Reviewer #2: No

---

## [Editor Report · Acceptance letter]

PONE-D-25-22924R1

PLOS One

Dear Dr. Shental,

I'm pleased to inform you that your manuscript has been deemed suitable for publication in PLOS One. Congratulations! Your manuscript is now being handed over to our production team.

Kind regards,

on behalf of

Dr. Theodore Raymond Muth

Academic Editor

PLOS One